# How Do Gepotidacin and Zoliflodacin Stabilize DNA-Cleavage Complexes with Bacterial Type IIA Topoisomerases? 2. A Single Moving Metal Mechanism

**DOI:** 10.3390/ijms26010033

**Published:** 2024-12-24

**Authors:** Robert A. Nicholls, Harry Morgan, Anna J. Warren, Simon E. Ward, Fei Long, Garib N. Murshudov, Dmitry Sutormin, Benjamin D. Bax

**Affiliations:** 1Scientific Computing Department, UKRI Science and Technology Facilities Council, Harwell Campus, Didcot, Oxfordshire OX11 0DE, UK; 2Medicines Discovery Institute, Cardiff University, Cardiff CF10 3AT, UK; 3Diamond Light Source, Harwell Campus, Didcot, Oxfordshire OX11 0DE, UK; 4MRC Laboratory of Molecular Biology, Cambridge CB2 0QH, UK; 5Institute for Systems Biology, Seattle, WA 98109, USA

**Keywords:** novel bacterial topoisomerse inhibitor (NBTI), spirocyclic pyrimidine trione (SPT), fluoroquinolone, moxifloxacin, gepotidacin, zoliflodacin, type IIA topoisomerase, DNA-cleavage

## Abstract

DNA gyrase is a bacterial type IIA topoisomerase that can create temporary double-stranded DNA breaks to regulate DNA topology and an archetypical target of antibiotics. The widely used quinolone class of drugs use a water–metal ion bridge in interacting with the GyrA subunit of DNA gyrase. Zoliflodacin sits in the same pocket as quinolones but interacts with the GyrB subunit and also stabilizes lethal double-stranded DNA breaks. Gepotidacin has been observed to sit on the twofold axis of the complex, midway between the two four-base-pair separated DNA-cleavage sites and has been observed to stabilize singe-stranded DNA breaks. Here, we use information from three crystal structures of complexes of *Staphlococcus aureus* DNA gyrase (one with a precursor of gepotidacin and one with the progenitor of zoliflodacin) to propose a simple single moving metal-ion-catalyzed DNA-cleavage mechanism. Our model explains why the catalytic tyrosine is in the tyrosinate (negatively charged) form for DNA cleavage. Movement of a single catalytic metal-ion (Mg^2+^ or Mn^2+^) guides water-mediated protonation and cleavage of the scissile phosphate, which is then accepted by the catalytic tyrosinate. Type IIA topoisomerases need to be able to rapidly cut the DNA when it becomes positively supercoiled (in front of replication forks and transcription bubbles) and we propose that the original purpose of the small Greek Key domain, common to all type IIA topoisomerases, was to allow access of the catalytic metal to the DNA-cleavage site. Although the proposed mechanism is consistent with published data, it is not proven and other mechanisms have been proposed. Finally, how such mechanisms can be experimentally distinguished is considered.

## 1. Introduction

One of the aims of modern structural biology is to understand how movement is coupled to the chemical making and breaking of bonds, that is to “watch chemistry happen” [1]. One of the long-standing challenges is elucidating the catalytic mechanism of topoisomerases—complex and flexible enzymes, which modify the topology of DNA. Topoisomerases are divided into two classes—type I and type II (see Appendix A). While type I topoisomerases use single-stranded DNA breaks to modify DNA topology, type II use double-stranded DNA breaks [2,3,4,5]. The introduction of double-stranded DNA breaks is potentially lethal to cells and is normally carefully controlled. Type IIA topoisomerases (Figure 1) include two human enzymes, TOP2A and TOP2B [5], as well as the bacterial enzymes topoisomerase IV (topo IV) and DNA gyrase. Interestingly, *M. tuberculosis* only has DNA gyrase (i.e., it lacks topo IV) and a type IA topoisomerase [6,7], which is mechanistically similar. Figure 1b, shows a simplified view of the function of a type IIA topoisomerase. A double-stranded break is made in one segment (the gate or G-DNA duplex) and another DNA duplex is passed through that break. This is accompanied by large movements of the enzyme [2,8,9].

How the two bacterial type IIA topoisomerases, DNA gyrase and topo IV cleave DNA and modulate DNA topology is of interest because three classes of antibiotics stabilizing DNA-cleavage complexes with these targets have now successfully passed phase III clinical trials. The flexibility these enzymes require, in making a double-strand DNA break in one DNA duplex and then passing a second DNA duplex through that break (Figure 1), has caused problems in obtaining high-resolution crystal structures. A clear definition of how the catalytic metal ion(s) regulate this DNA-cleavage process in a normally safe manner follows from high-resolution (better than 2.2 Å) crystal structures [10]. The deletion of the small Greek Key domain in *S. aureus* DNA gyrase improved the resolution of crystal structures with GSK299423 (a precursor of gepotidacin) from 3.5 Å (with the Greek Key domain) to 2.1 Å (with the Greek Key domain deleted) and gave a clear view of a catalytic metal at the 3′(A) position [11]. This *S. aureus* DNA gyrase Greek Key deletion construct has been successfully used in a large number of crystal structures [10,12], but has not yet been successfully replicated in other type IIA topoisomerases, which all contain the small Greek Key domain.

An initial structure of moxifloxacin with *A. baumanii* topo IV in a DNA-cleavage complex [13] showed the presence of the water–metal ion bridge [14] and explained two common target-mediated resistance mutations. However, the DNA used in this initial structure was from a previous lower-resolution structure [15] and the limited resolution of the data (3.25 Å) precluded a clear visualization of the register of the DNA [13]. Subsequent higher-resolution structures with a shorter version of the DNA, quinolones and a *M. tuberculosis* DNA gyrase construct [16] clearly showed that the asymmetric DNA sequence was in two orientations in the crystal—related by the twofold axis of the complex. The register of the DNA sequence in the initial *A. baumanii* topo IV structure is uncertain [13]. A 2.95 Å structure of the *S. aureus* DNA gyrase Greek Key deletion construct (Appendix A) in a DNA-cleavage complex with moxifloxacin (PDB code: 5cdq [17]), used a 20-base-pair symmetric (palindromic) DNA sequence to avoid the possibility of the DNA suffering from static disorder around the twofold axis of the complex [18]. This moxifloxacin structure (5cdq-BA-x.pdb—Appendix A) is very similar to the moxifloxacin DNA-cleavage complexes with *A. baumannii* topo IV [13]. However, the 2.95 Å structure is with the Greek Key deletion mutant (see Appendix A). The water–metal ion bridge [14] is conserved in moxifloxacin DNA-cleavage complex structures between *A. baumanii* topo IV [13] and *S. aureus* DNA gyrase [17] and in a modified form in *M. tuberculosis* DNA gyrase [16], (see Figure 3 in [16], and note the presence of an alanine at position 90 in *M. tuberculosis* GyrA—cf. serine in other bacteria). Interestingly the *Mycobacterium tuberculosis* ‘*genome encodes only one copy of type I and one copy of type II topoisomerase*’ [6].

**Figure 1 ijms-26-00033-f001:**
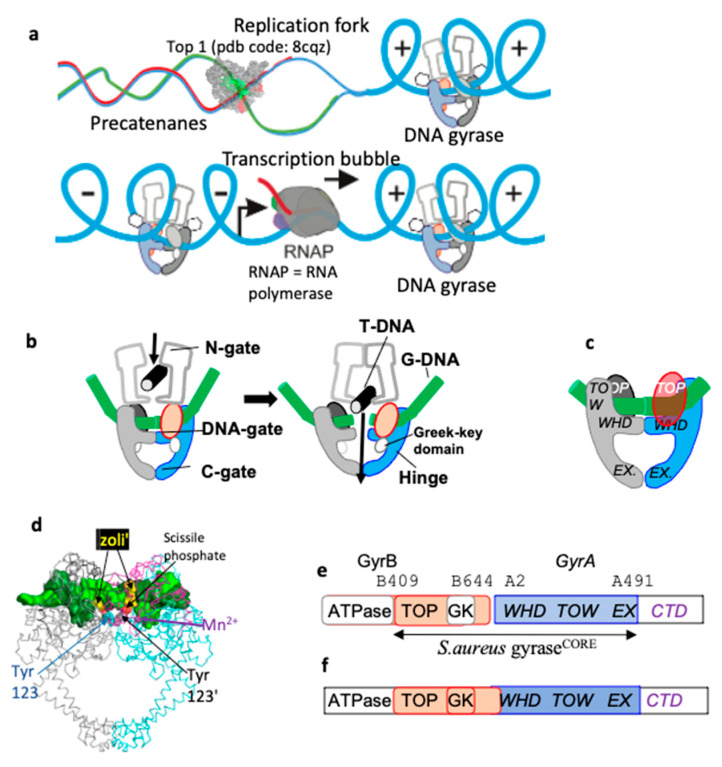
***S. aureus* DNA gyrase is a type IIA topoisomerase.** (**a**) Type IIA topoisomerases work ahead of replication forks (and transcription bubbles, at least for long genes [5]) to remove positive DNA supercoils [19,20,21,22] to allow both DNA replication (and DNA transcription) to take place; in *M. tb* Top1 may relieve negative supercoils before Okazaki fragments are ligated [23]. Proteins (DNA polymerase, helicases and primases) working at the replication fork are not shown. (**b**) Simplified schematic of ATP-dependent relaxation of negatively supercoiled DNA carried out by a type IIA topoisomerase. The gate or G-DNA (green cylinder) is cleaved and another DNA duplex, the T (or transport segment—black) is passed through the cleaved DNA before religation. (**c**) A schematic of *S. aureus* DNA gyrase GyrB27–A56(GKdel) DNA complex with uncleaved DNA (green). One fusion truncate is shown in red and blue, the other in grey and black. Note the small Greek Key domain (residues 544–579) in the GyrB subunit has been deleted and replaced with two amino-acids. (**d**) View of a 2.8 Å crystal structure of a GyrB27–A56(GKdel) complex with zoliflodacin (zoli’) and doubly cleaved DNA (PDB code: 8bp2). The protein is shown as a Cα trace with GyrA subunits cyan or grey and GyrB subunits magenta or black; the cleaved DNA (surface representation) is colored green or forest green (a similar coloring scheme is used throughout this paper). The side chains of the catalytic tyrosines, which are covalently bonded to the cleaved DNA via the scissile phosphates, are shown in sphere representation GyrA Tyr 123 (cyan) and Tyr 123′ (grey). Zoliflodacins are shown as spheres (yellow). An Mn^2+^ ion at the Y (or B) site is shown as a purple sphere. (**e**) DNA gyrase consists of two subunits, GyrB and GyrA. The *S. aureus* DNA gyrase GyrB27–A56(GKdel) construct used to determine many crystal structures is a fusion of the C-terminal Toprim (TOP) domain of GyrB with the winged helical domain (WHD), tower (TOW) and exit-gate (EX) domains from GyrA (colored region (**e**,**f**); in (**f**) the Greek key domain is also colored). The C-terminal domain (CTD), which wraps DNA in DNA gyrase, is not shown in panel b (the CTD has different functions in other type IIA topoisomerases). (**f**) In humans (and yeast), topoII is a single subunit and functions as a homodimer.

An extensive corpus of biochemical and structural information strongly suggests that type IIA topoisomerases require a catalytic metal (preferentially Mg^2+^) in the Toprim domain for proper DNA cleavage and religation; while type IA topoisomerases seem to only require the metal ion for religation by the Toprim domain [7], DNA cleavage is accomplished with the aid of a lysine residue. Type IA topoisomerases only work on negatively supercoiled DNA [7]. There are currently three theories (see Table 4 and discussion in [10]) of how the G-segment DNA is cleaved by type IIA topoisomerases such as DNA gyrase: (i) A two-metal mechanism that was proposed [24] based on a misinterpreted electron density map. To date, to the best of our knowledge, there has been no structural data published that are consistent with such a model, and therefore it is unlikely to be correct. (ii) A single-metal mechanism in which DNA cleavage is proposed to take place when the metal ion is at the 3′(A) site, as described in [12] (pp. 3438–3447). This model is also unlikely to be correct [25,26], noting that it does not explain why the catalytic tyrosine is in the tyrosinate (negatively charged) form, nor explain how the enzyme uses supercoiling to control DNA cleavage. And (iii) the single moving metal mechanism described in this paper, which explains why the catalytic tyrosine is in a tyrosinate (negatively charged) form for DNA cleavage, and proposes that the small Greek Key domain controls metal access to the catalytic site to allow rapid relaxation of positively supercoiled DNA ahead of replication forks and transcription bubbles (see Figure 1). In this paper, a chemically sensible single moving metal mechanism [25,26] is proposed, based on three published crystal structures [11,17,27]. The aim of this paper is not to prove a mechanism, but to present a new hypothesis consistent with existing data (including chemistry). Experiments are then proposed to “watch chemistry happen”.

Metal-binding Toprim domains were named for topoisomerases and primases and were characterized in a multiple sequence alignment in 1998 [28]. However, the first Toprim domain in a type IIA topoisomerase was described in the structure of the catalytic core of yeast topoisomerase II published in 1996 [29]. This stated that in addition to the Toprim domain a simple Greek Key domain ‘(*residues 565–605*) *is inserted between the fifth strand and final helix* [of the Toprim domain]’. Whilst Toprim domains are found in many other proteins, including other topoisomerases (e.g., type IA topoisomerases), the insertion of the simple Greek Key domain between the fifth strand and subsequent helix of the Toprim domain seems to be a feature that is found uniquely in all type IIA topoisomerases. In Gram-negative DNA gyrases (such as *E. coli* DNA gyrase), the Greek Key domain is augmented by an inserted domain of some 174 residues; this inserted domain is not present in other type IIA topoisomerases, which have only the small Greek Key domain inserted in the Toprim domain (see the sequence alignment in Appendix A). The small Greek Key domain in type IIA topoisomerases tends to be mobile—it was not seen in the 2.15 Å structure of etoposide with human TOP2B [30].

A *S. aureus* DNA gyrase fusion construct (Figure 1) in which the small Greek Key domain was deleted (residues 544–579 replaced with two amino acids) [11], allowed a clear visualization of a 3′(A) site metal (with a YtoF mutant and GSK299423). A long-standing question has been why type IIA topoisomerases have the additional Greek Key domain within the Toprim domain. One of the conclusions of this paper is that the Greek Key domains within type IIA topoisomerases first arose to allow control of metal access to the active site—specifically allowing rapid access of metal ions to control relaxation of positively supercoiled DNA in front of replication forks [19,20,21,22].

## 2. Results

In developing the mechanism/movie proposed in this paper, we used four complexes from three crystal structures (see Appendix A): (i) the 2.1 Å ternary complex with GSK299423 (a precursor of gepotidacin) and uncleaved DNA (PDB code: 2xcs, [11]), which has the catalytic tyrosine mutated to phenylalanine and gives a clear view of a 3′(A) site metal ion (see Table 1); (ii) a 2.5 Å DNA-cleavage complex stabilized by QPT-1 (PDB code: 5cdm [17]), the progenitor of zoliflodacin (Figure 2c), which has a single metal ion observed at the Y(B) site; and (iii) the two complexes present in the asymmetric unit of the 2.6 Å binary complex (PDB code: 6fqv). Models corresponding to these two complexes (6fqv-c1.pdb and 6fqv-c2.pdb) are available from the ‘Research’ tab at https://profiles.cardiff.ac.uk/staff/baxb, accessed on 11 December 2024. These complexes are similar apart from the position of the catalytic tyrosine (see Figure 2). Neither complex has a metal ion at an active site [27].

Before a movie could be made, the four sets of crystallographic coordinates (see Table 1) had to be adjusted so that: (i) The structural models had the same DNA sequences; (ii) each set of coordinates contained a single molecule (electron density maps typically represent thousands of molecules in the crystal, so side chains where there was electron density for more than one conformation, or no visible supporting electron density, were modelled as single conformations); (iii) hydrogen atoms were added to structures using the program Maestro [31] (see Section 4 for details).

**Figure 2 ijms-26-00033-f002:**
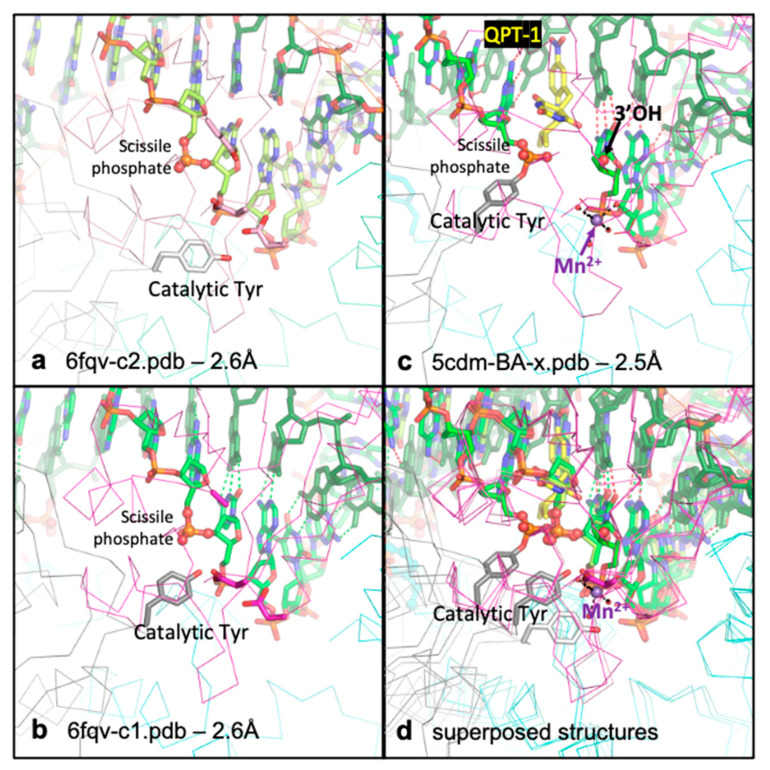
**The catalytic tyrosine appears to move.** (**a**) View of binary complex 2 (6fqv-c2.pdb. Note that the conformation at the second active site of this—and other—complexes is similar to that at the first active site). (**b**) View of binary complex 1 (6fqv-c1.pdb). (**c**) Equivalent view of 2.5 Å QPT-1 complex (5cdm-BA-x.pdb). Note in this structure the catalytic tyrosine has cleaved the scissile phosphate and the QPT-1 sterically inhibits the approach of the scissile phosphate to the 3′OH for religation of the DNA. The metal ion (Mn^2+^) is at the Y (B) site. (**d**) The three structures, from panels a, b and c are shown superposed. Note the relative positions of the catalytic tyrosine—suggestive of its movement. Structural figures in this paper were created using PyMOL, v1.8.4.0 [32].

### 2.1. The Catalytic Tyrosine Seems to Move

To model chemistry happening, based on multiple crystal structures (Table 1), it is useful if the catalytic metal ion has the same name in different crystal structures. As such, the standard BA-x nomenclature is used for *S. aureus* gyrase PDB files (see Section 4 and [10] for details of this naming system). The catalytic tyrosine is residue 123 in *S. aureus* DNA gyrase = 122 in *E. coli* DNA gyrase (see Appendix A).

In Figure 2, the position of the catalytic tyrosine is shown in three different positions, in three different complexes. In 6fqv-c2.pdb (Figure 2a) the catalytic tyrosine (from the second complex in the asymmetric unit from 6fqv.pdb) is not in the catalytic pocket. Whereas in 6fqv-c1.pdb (Figure 2b), that catalytic tyrosine has a position where it can displace a water molecule from the normal coordination sphere of a Y(B) site metal. However, no metal ions are observed in the catalytic pocket in either complex in the 6fqv crystal structure [27]. This seems to be because the phosphate prior to the scissile phosphate is too far away from the metal binding site to interact with a water. In high-resolution crystal structures where a Y(B) site metal is observed (e.g., [30]), a water molecule is observed sitting between the catalytic metal ion and the phosphate prior to the scissile phosphate (from the DNA). In this paper, the Y(B), 3′(A) nomenclature is used for the two observed metal positions (see [10] for details). This is because, as shown in Figure 2, the catalytic tyrosine appears to move.

When the catalytic tyrosine cleaves the DNA (as shown in Figure 2c), it becomes covalently attached to the scissile phosphate. Note that the active sites are formed in trans with the catalytic tyrosine C123′ from the C subunit (of the DC-fusion protein) cleaving the DNA with the B-subunit (from the BA-fusion protein) Toprim domain (and, vice versa, tyrosine A123 cleaving with the D-subunit Toprim domain). Standard crystallographic nomenclature—where residues from the first (BA) chain are not given a prime (′), whereas residues from the second (DC) chain are (e.g., C123′ or just 123′)—is used below in this paper.

### 2.2. A Moving Metal Mechanism for DNA Cleavage

The position of the catalytic tyrosine in 6fqv-v1.pdb is such that its terminal oxygen is within 1.6 Å of water 5095 in 5cdm (see Figure 2 and Appendix A), i.e., the terminal oxygen of the tyrosine and water 5095 occupy a similar position in space. Because the catalytic tyrosine appears to move (see Figure 2), the terminal oxygen of the tyrosine can be moved about 1.6 Å back towards its position in 6fqv-c2.pdb (Figure 2) to be closer to the position of water 5095—and displace it from the Mn^2+^ coordination sphere (see Section 4; Table 2 and Table 3). In this starting position, the tyrosine is accepting a hydrogen bond from Arg 122′, but the slight move backwards means Arg 122′ is too far from the scissile phosphate to donate a hydrogen bond to the scissile phosphate. The catalytic tyrosine (A123 or C123′) is believed to be in the tyrosinate (O- rather than OH) form before it cleaves the DNA (see Section 3). In cleaving the DNA, the tyrosinate form is stabilized by interactions with both the catalytic metal (Mn^2+^) and the positively charged side chain of Arg122 (Figure 3a). Note that Arg A122 (or Arg C122′) is an important catalytic residue but is often quite mobile in *S. aureus* gyrase crystal structures.

The model presumes that the metal comes in as the catalytic tyrosine moves. One possibility is that the metal access to the active site may be controlled by the Greek Key domains for positively supercoiled DNA (see Section 5).

In the proposed DNA-cleavage mechanism, the Y(B) site metal is coordinated by the catalytic tyrosine (Y) prior to DNA cleavage (Figure 3a) and the metal is then attracted towards the 3′(A) site (Figure 3b) but the bond between the 3′ oxygen and the phosphorous atom of the scissile phosphate is cleaved just before the metal reaches the 3′(A) site (Figure 4). Figure 4b suggests that the hydrogen atom from water 5093 is, when transferred to the 3′ oxygen, between this oxygen and the scissile phosphate. However, we now believe the hydrogen on the 3′ oxygen is attracted towards Glu B435 and a hydrogen bond forms between this 3′OH and Glu B435 on DNA cleavage (see Appendix A).

Note in the binary complex, 6fqv, the side chain of Glu 435 is disordered; presumably it tries to move away from the scissile phosphate and the side chains of Asp 508 and Asp 510. In structures containing a metal ion in the active site, Glu 435 becomes ordered. In the DNA-cleavage mechanism (Figure 4), on entering the active site the metal ion (Mg^2+^ or Mn^2+^) is attracted towards two negative charges, on the scissile phosphate and on Glu 435, and as it moves it maintains its contact with the side chain of Asp 508 (Figure 3c). Water 5093, which protonates the 3′ oxygen to effect DNA cleavage, moves as the metal ion moves (Mn^2+^—purple sphere—Figure 4a).

After the DNA has been cleaved (Figure 4), the DNA gate can be pushed open by a traversing T-DNA duplex, while the metal ion moves back to the Y(B) site, where it acquires a new water—water 5095—replacing the tyrosine oxygen in the metal ion coordination sphere (see Appendix A). Once the T-DNA has passed through the doubly cleaved DNA, presumably the metal at the B site can religate the DNA.

It is possible, in principle, to propose a simple religation scheme, where water 5095 moves to protonate the tyrosine oxygen, regenerating the tyrosine from the phosphotyrosine. The metaphosphate-like intermediate is presumably then attacked by the 3′ O^−^ ion to regenerate the intact DNA. However, such a religation mechanism has some complications. The 3′OH probably loses a proton to Glu 435, before becoming the attacking 3′ O^−^ ion, but more problematical is the exact conformation of the phosphotyrosine when it is protonated by water 5095. We have not yet fully elucidated such a scheme.

## 3. Discussion

The scheme for DNA cleavage shown in Figure 4 suggests that the initial step in the cleavage of the DNA is the protonation of the 3′ oxygen. This contrasts with many DNA-cleavage mechanisms which propose that the initial step is caused by a tyrosinate ion attacking a tetrahedral negatively charged phosphate (e.g., Figure 4c in a paper on type IA topoisomerases [33]). The nucleophilic attack of a negative tyrosinate ion on a negatively charged phosphate ion seems chemically unlikely. The mechanism proposed in this paper for *S. aureus* DNA gyrase suggests that (Figure 4 and Figure 5) the DNA is first cleaved by protonation of the negatively charged scissile phosphate and then the extremely reactive metaphosphate-like planer intermediate is attacked by the catalytic tyrosinate (Figure 5c). The mechanism proposed in this paper is informed by the non-enzymatic hydrolysis of methylphosphate mechanism (Figure 10.2 and Scheme 10.1 in Frey and Hegeman [34]). However, exactly how the protons (hydrogens) move during catalysis is not yet clear (see below for details). Previous mechanisms proposed for type IIA topoisomerases include a mechanism based on Figure 7 (entitled ‘Dynamic model for the cleavage religation cycle catalyzed by a single metal ion’) from Bax et al. (2019) [12] that lacks sensible chemistry—we assert that this mechanism seems unlikely to be correct (see Table 4 in the accompanying paper [10] for ‘a comparison of models proposed in literature for the first DNA cleavage step of a type IIA topoisomerase’). Similarly, a proposed ‘two metal mechanism’ [24] seems to be based on the misinterpretation of an electron density map [10]. This structure has been re-refined (RR-3l4k.pdb) to have chemically reasonable geometry and a single metal ion at each active site, consistent with single moving metal mechanisms [12]. As no other structures have yet been produced that are consistent with this mechanism (to the best of our knowledge), this also seems unlikely to be correct. Although it is difficult to disprove mechanisms [35], lack of contradictory evidence does not imply they are true. We assert that proposed mechanisms should be consistent with the current knowledge of metal ion coordination geometry [25,26].

**Figure 5 ijms-26-00033-f005:**
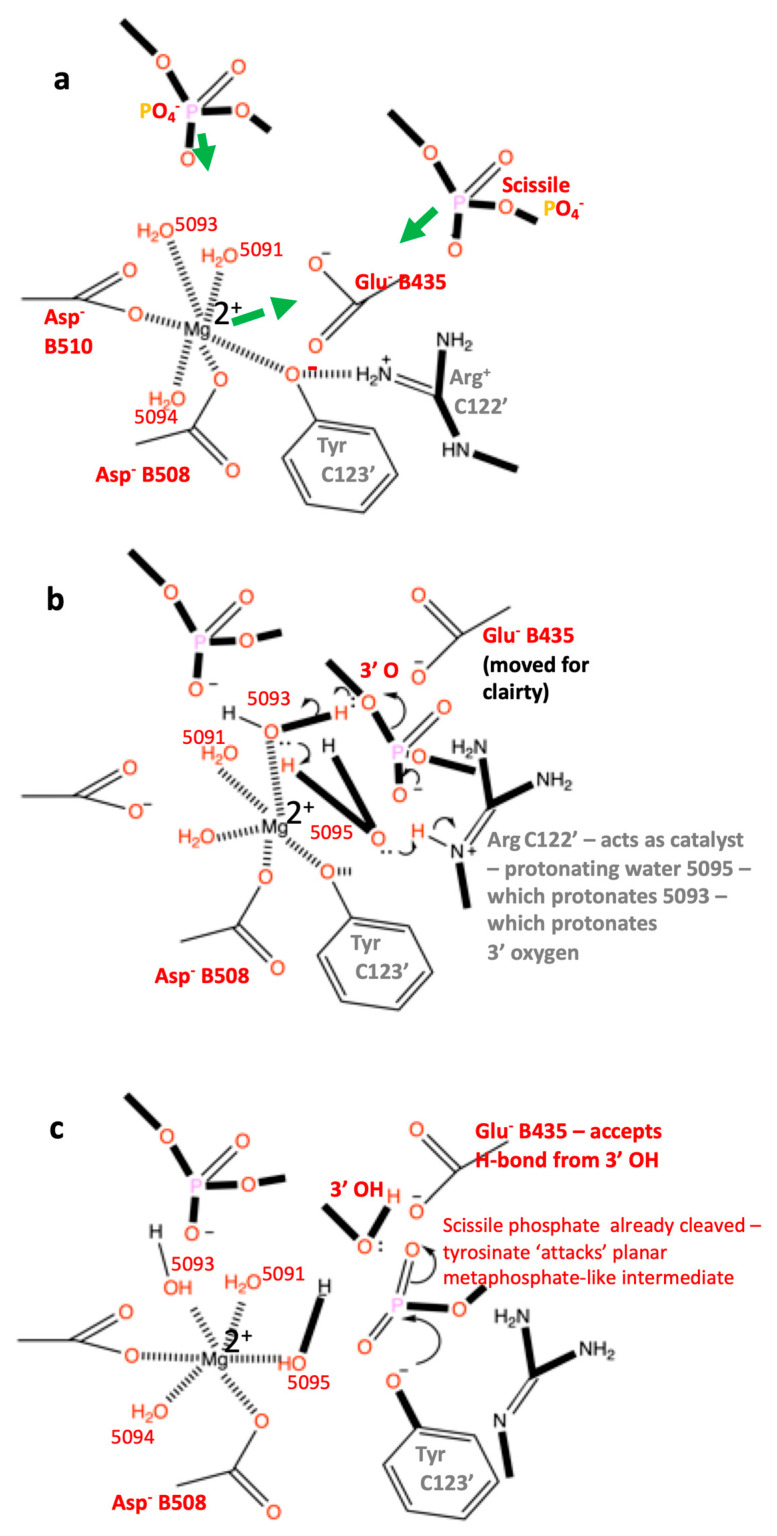
**A mechanism for DNA cleavage by *S.*
*aureus* DNA gyrase.** (**a**) The catalytic metal, Mg^2+^, comes into the active site pocket with the catalytic tyrosine (C123′). Because the side chain oxygen is accepting a hydrogen bond from Arg 122′ and is also coordinating the e Mg^2+^ ion it is in the tyrosinate (negative ion) form already. The PO_4_^−^ ion prior to the scissile phosphate moves down to accept a hydrogen bond from water 5093. The Mg^2+^ ion is attracted towards Glu B435 and the scissile phosphate, and these are attracted towards the Mg^2+^ ion. (**b**) The Mg^2+^ ion has moved towards the A site—this causes water 5093 to move and protonate the 3′ oxygen from the scissile phosphate, which then becomes the 3′ OH. However, almost simultaneously water 5095 comes into a new ‘hole’ and is immediately protonated by the N-epsilon hydrogen from Arg 122′. The scissile phosphate becomes a highly reactive planar metaphosphate-like group. (**c**) The catalytic tyrosinate ion, C123′, attacks the planar metaphosphate-like intermediate to generate the phosphotyrosine. Note Arg C122′ still needs to pick up a proton to regain its +1 charge. The catalytic metal returns to the Y(B) site with water 5095. In the presence of a T-DNA, the cleaved G-DNA gate will open to allow T-DNA passage, before religation of the gate DNA. (Figure drawn with Chemdraw [36]).

One mechanism proposed in this paper (Figure 5) starts with protonation of the negatively charged DNA backbone by a water molecule, which is in turn protonated by a second water molecule (Figure 5b). The highly reactive metaphosphate-like intermediate is then immediately attacked by the negatively charged tyrosinate ion (Figure 5c). This mechanism seems chemically more sensible than the conventional mechanism that starts with the tyrosinate ion making a nucleophilic attack on a negatively charged tetrahedral phosphate from the DNA, ending with protonation of the 3′ oxygen (see for example the mechanism discussed in [7]). Although the proposed mechanism seems chemically more sensible, it is not yet proven and depends on hydrogen atom transfer. If there is no tyrosinate positioned to attack then the metaphosphate-like intermediate will presumably immediately re-attack the 3′-OH, regenerating the uncleaved DNA. In an alternative mechanism (Appendix A), water 5093 could be directly protonated by Arg 122′ to affect the DNA cleavage. Such a mechanism (Appendix A) suggests that the single-stranded DNA cleavage seen with NBTIs such as GSK299423 [11] and gepotidacin [37] is likely due to incomplete religation at one of the two active sites. A structure of the E435Q mutant might prove this mechanism; the metal ion should enter the Y(B) site and stay there. Arg 122 should be attracted to interact with the scissile phosphate, but then be stuck as the metal ion remains at the Y(B) site (not being attracted towards E435Q).

Seeing such chemistry happen will require carefully designed experiments where hydrogen atoms could be observed, such as by neutron [38] or electron diffraction experiments. Advancements over the last 10 years in electron diffraction show that this technique has potential to better locate hydrogen atoms, and could in the future be used to understand this mechanism further [39]. Experimental definition of hydrogen atom positions is challenging, and perhaps this explains why QM/MM studies [40] are still used despite their obvious limitations [41,42]. A more definitive answer awaits the development of experimental techniques that allow hydrogens to be clearly visualized in multiple macromolecular structures.

We believe that the most likely current explanation of the observed synergy [24,43,44,45] between different metal ions in type IIA topoisomerases is that when one active site contains a Y(B) site metal the other uses a 3′(A) site metal. Presumably differences in the ionic radius of calcium [25] account for calcium’s tendency to give more DNA cleavage in the absence of a compound [46,47]. Similarities between type IA and type IIA topoisomerases, which both have Toprim domains, suggest that they likely have similar mechanisms [7]. Although type IA topoisomerases only work on negatively supercoiled DNA and can cleave a single-stranded DNA segment in the absence of a divalent metal ion [7], they require a divalent metal ion for DNA-religation [7,48]. Dual targeting of DNA gyrase and topo IV [49] is important to prevent the likelihood of developing resistance to drugs. If a compound could be found that inhibits both *M. tb* topo I and DNA gyrase (Appendix A), it would be interesting to measure the likelihood of resistance to such a compound.

## 4. Materials and Methods

The aim of this paper is to present a new moving metal mechanism that is consistent with existing crystal structures. The mechanism depends on the accuracy of the metal-ion coordination geometry and re-refined coordinates are available [10] for the two crystal structures containing catalytic metal ions (2xcs-v2-BA-x.pdb and 5cdm-v2-BA-x.pdb). However, we note that crystal structures are static pictures of thousands of molecules that make up a crystal.

What we are concerned with in this paper is how DNA-gyrase cleaves DNA. As such, we used the following procedure: (i) Coordinates from the three complexes with the same chain IDs (6fqv-c1.pdb, 2xcs-v2-BA-x.pdb and 5cdm-v2-BA-x.pdb) were manually edited in coot [50]—so that each coordinate set contained a complete set of single atoms. Also, the DNA sequence was manually edited so that it was the same in the three starting structures (see Table 2 for details). This gave: 6fqv-c1-one-DNA.pdb, 2xcs-v2-one-DNA.pdb and 5cdm-v2-one-DNA.pdb. Note 6fqv-c1-one-DNA.pdb contains no catalytic metal ion and Glu B435 was modelled in in a conformation similar to that observed in 5cdm. The two sets of coordinates with a metal ion, 2xcs-v2-one-DNA.pdb and 5cdm-v2-one-DNA.pdb, were read into Maestro [31] (ii) The hydrogens were added onto all atoms in ‘prepare’ [51]. The structure was written out from Maestro (export structure) and edited so the header contained a CRYST1 record. Then, because the hydrogen atoms placed on waters were all pointing in the same direction, the relevant waters were manually rotated (in coot [50]) around the oxygen atom to chemically sensible positions [52]. The coordinates with manually rotated waters were then read back into Maestro [31]; for relevant waters, just the hydrogens were selected and then energy-minimized in Maestro (‘minimize selected atoms’). The manganese atom had a formal charge of +2 in Maestro. We used this procedure to keep heavy atom coordinates (non-hydrogen atoms) used for making movies close to those experimentally defined from crystal structures [10]. This gave 2xcs-v2-two-DNA.pdb and 5cdm-v2-two-DNA.pdb, as explained in Table 2; and with 6fqv-cd.pdb and 6fqv-c1-one.pdb, these were used to help guide the making of the molecular DNA-cleavage movie.

To aid visualization of our proposed mechanism of DNA cleavage in type IIA topoisomerases, we created a molecular movie that morphs between four states (Table 3): (i) uncleaved DNA with no metal ion present; (ii) entry of the metal ion coordinated by the catalytic tyrosine; (iii) transient state at the point of DNA cleavage; and (iv) cleaved DNA with the metal ion still bound. The movie was produced using the ProSMART library for structure analysis [53], similarly as in [54]. The procedure involves applying hierarchical aggregate transformations to residues, backbone and side chain atoms, resulting in a parsimonious morphing between states. The resultant ensemble was rendered into a movie using PyMOL [32].

This procedure required preparation of coordinate models corresponding to four states (see Table 3):

State 1—uncleaved DNA (6fqv-c2): Due to this model being the second complex in 6fqv, the chain ID nomenclature is different due to the requirement for uniqueness. Specifically, the protein chains named R/S/T/U (authors’ original annotation) correspond to the conventionally named chains A/B/C/D, and similarly the nucleic acid chains V/W correspond to E/F. To achieve an equivalent coordinate frame for producing the movie, the complex was superposed onto the other complex, aligning chain S in 6fqv-c2 to chain B in 6fqv-c1 using ProSMART [53]. In 6fqv-c2, Arg122 is modelled in two conformations; conformer A was removed and B retained, so as to retain the hydrogen bond between the Tyr C123 terminal oxygen and Arg C122(NH2) (an interaction that is relevant to the proposed mechanism).

State 2—entry of the metal ion (6fqv-c1): Despite the metal ion not being present in 6fqv, we transplanted the Mn^2+^ and coordinating waters from 5cdm, keeping the metal ion coordinated to Asp B508(OD2) and Asp B510(OD2). We manually adjusted the positions of the coordinating waters (B5091, B5093 and B5094) to ensure reasonable coordination geometry, with hydrogen positions optimized using Maestro [31]. Model 5cdm also has water B5095 coordinating the metal. However, we propose that the metal is brought into the binding site by Tyr C123, and that the C123 side chain oxygen atom takes the place of the water B5095 observed in 5cdm. Consequently, we manually adjusted the position of Tyr C123 (by rotating the side chain) so that the C123 side chain oxygen atom coordinates the metal ion in place of B5095. This resulted in the C123 side chain oxygen atom moving a distance of approximately 1.3 Å. Similarly, we manually adjusted the adjacent Arg C122 so as to maintain the hydrogen bond between C122(NH2) and side chain oxygen atom of C123. These ‘manual’ adjustments were applied using Coot [50], attempting to ensure reasonable geometry for the adjusted residues.

State 3—transient state at the point of DNA cleavage (between 6fqv-c1 and 2xcs): The model 2xcs exhibits uncleaved DNA, with the catalytic tyrosine C123 mutated to phenylalanine, and the metal ion coordinated by the OP1 atom of the uncleaved phosphate of DG E9. Our proposed mechanism involves the metal ion inducing cleavage prior to the structure entering the state observed in 2xcs, i.e., it is proposed that the metal ion does not quite reach the scissile phosphate. However, we do not have access to an intermediate state at the exact point at which cleavage occurs; it is clear why it would be difficult to observe such a transient state using conventional crystallographic methods. Consequently, it was necessary to generate an intermediate between the previous state (derived from 6fqv-c1) and 2xcs. This, State 3, was generated by morphing between State 2 and 2xcs (using ProSMART, as for the main movie), and extracting an intermediate set of coordinates at the point at which (we hypothesize) the DNA-cleavage-inducing chemical reaction occurs.

We also hypothesize that a water molecule moves into the binding site, to a position (reasonably close to water E52 in 2xcs) that is between Arg C122, Tyr C123, water B5093 and the phosphate of DG E9. We manually placed this water to a position that we believed most chemically sensible. In the movie, this water is seen to move in through an open pocket just prior to DNA cleavage. This water (named B5095 in Figure 5 and Appendix A, which is equivalent to water E52 in Figure 3) is protonated by Arg 122′ in Figure 5 and the movie. However, as can be seen in Figure 3, if Arg 122′ moves more slowly, another possibility is that Arg 122′ directly protonates water B5093 (see Appendix A).

State 4—cleaved DNA (5cdm): The final state is that with cleaved DNA as observed in 5cdm.

## 5. Conclusions

In 1979, Brown and Cozzarelli [55] proposed a sign inversion mechanism to explain the observed activity of *E.coli* DNA gyrase. They proposed that ‘*a positive supercoil is directly inverted into a negative one via a transient doublestrand break*’ and that the ‘*reverse process is the path by which gyrase, in the absense of ATP, relaxes negatively supercoiled DNA*’. A widely accepted model is that gyrase transfers a T-segment in a top-to-bottom movement through the temporary break in a G-segment to convert a right-handed turn formed by DNA wrapping around its CTD domain (a positive supercoil) into a left-handed turn with a T-segment below (a negative supercoil) [29]. However, an alternative model is topologically possible, with a right-handed turn formed by G and T-segments with a T-segment below the G-segment (a positive supercoil) converted to a left-handed turn (a negative supercoil) in a bottom-to-top movement of the T-segment (Figure 6). If this is the case, then the small Greek Key domains (Figure 7) would be ideally placed to recognize the incoming T-DNA segments and regulate access of metal ions to the active sites for fast relaxation of positively supercoiled DNA. Despite being topologically feasible, the authors did not meet a consensus regarding the plausibility of this alternative model, as it contradicts recent structural information that supports the existence of a right-handed turn with a T-segment above the G-segment, thus favoring the conventional model [8,56]. Consequently, the alternative model requires further investigative elaboration.

For bacterial inhibitors of type IIA topoisomerases, Appendix A suggests that NBTIs such as gepotidacin may promote single-stranded DNA cleavage—not by blocking the cleavage of one of the DNA strands, but rather by inhibiting religation of one of the two DNA strands. To the best of our knowledge, sequencing of the single-stranded DNA-cleavage sites, stabilized by NBTIs, has not yet been performed. If the complexes stabilized at 80% of sites with NBTIs such as gepotidacin contain no DNA cleavage, and only 20% of the sites contain single-stranded DNA cleavage, exactly how this would be distinguished in DNA-cleavage experiments remains unclear.

**Figure 7 ijms-26-00033-f007:**
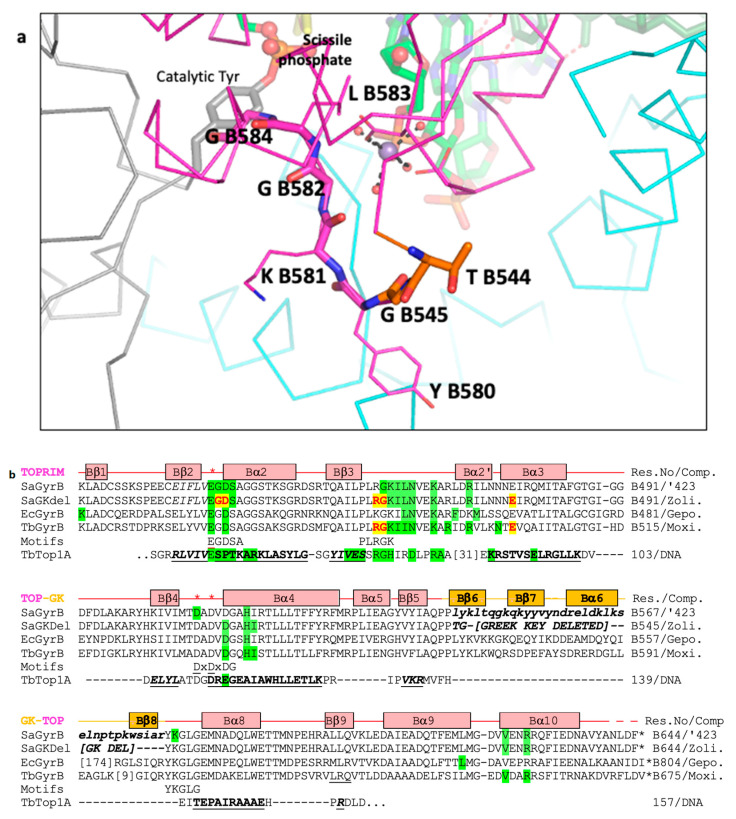
**The Greek Key deletion and YKGLG motif are close to the catalytic site.** (**a**) A view of the position of the two residues, T B544 and G B545 (orange carbons), that have replaced the Greek Key domain in 5cdm-BA-x.pdb. Note that the Greek Key motif, named after a pattern that was common on Greek pottery, normally contains four β-strands connected by hairpins. However, in type IIA topoisomerases the third β-strand of the Greek Key is replaced by an α-helix, and insertions in Gram-negative DNA gyrases are found in the connection to the final β-strand. Because the final β-strand is adjacent to the first β-strand it has been possible to delete the Greek Key domain in *S. aureus* DNA gyrase. (**b**) An alignment (initially with Clustal W [57]) of the amino acid sequences [58] from four Gyrase structures: *S.aureus* GyrB + GSK299423 (SaGyrB 3.5 Å—PDB code: 2xcr), *S. aureus* GyrB + zoliflodacin (SaGKdel 2.8 Å—PDB code: 8bp2), *E. coli* GyrB + gepotidacin (EcGyrB 4.0 Å—PDB code: 6rks), and *M. tuberculosis* GyrB + moxifloxacin (TbGyrB 2.4 Å—PDB code: 5bs8). The *S. aureus* structures are crystal structures of the catalytic core (see Figure 1). The *E. coli* structure is a cryo-EM structure containing full length GyrA and GyrB. The motifs line highlights four conserved sequence motifs, three of which—(i) EGDSA, (ii) DxDxDG and (iii) the YKGLG motif (after the Greek Key domain)—are close to the catalytic metal (red asterisks above underlined catalytic residues). Amino acids are highlighted on the sequences if they contact (<3.8 Å) either the compound (red letters on yellow highlight), or the DNA (green highlight; note zoliflodacin GD and RG residues also contact the DNA; only the G from the RG motif also contacts the DNA in the moxifloxacin structure). The *M. tuberculosis* Top1A (TbTop1A) sequence is from the 2.78 Å structure with T and G-segment DNA (PDB code: 8czq) [23]. For TbTOP1 only the Toprim domain sequence is shown with secondary structural elements underlined in bold. The TOP-GK line shows the secondary structural elements from the 3.5 Å SaGyrB structure with GSK299423. Numbers in brackets indicate extra residues not shown (specifically: [31], [174] and [9]) (see Figure S5 in [17] for contacts on GyrA).

## Figures and Tables

**Figure 3 ijms-26-00033-f003:**
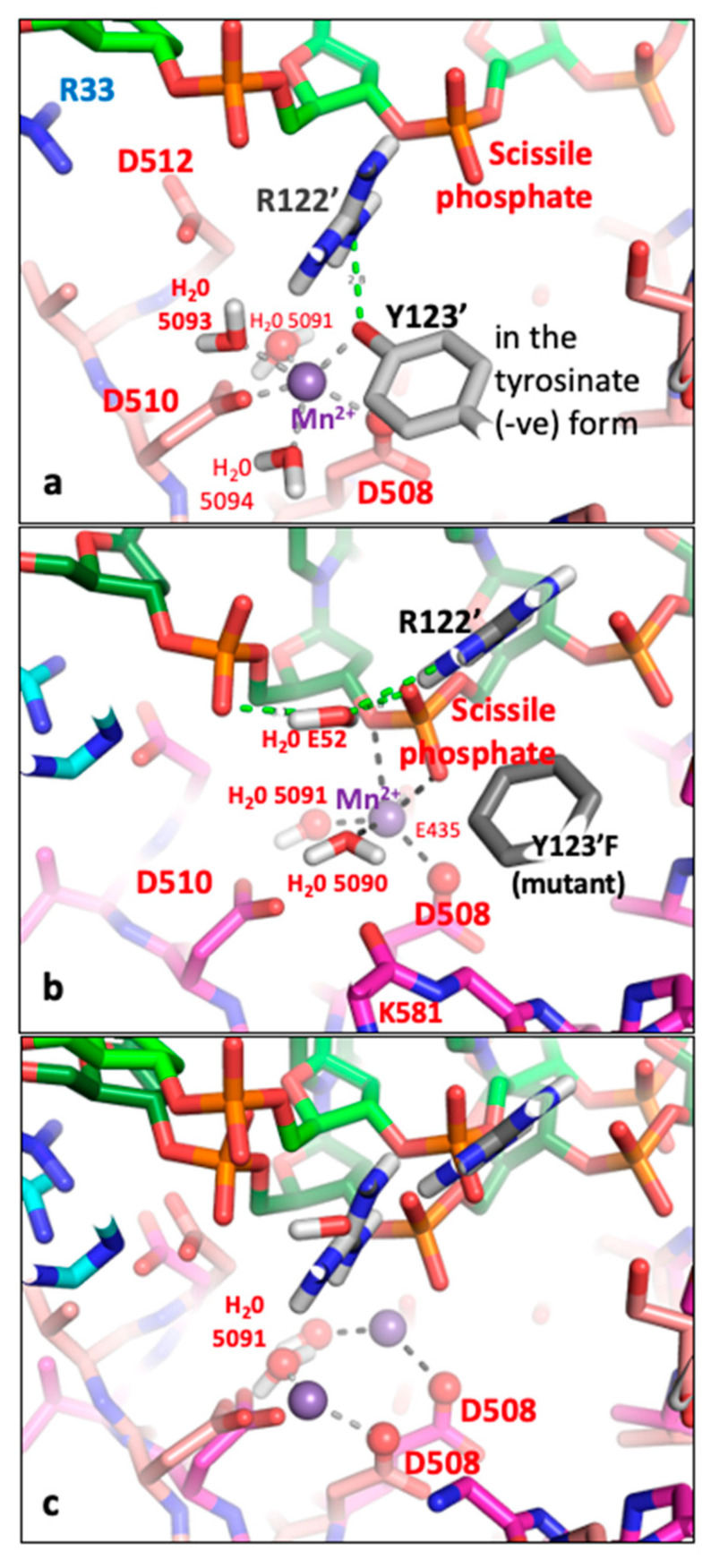
**A comparison of state2.pdb with 2xcs-v2-two-DNA.pdb—only some hydrogens (white atoms) are shown.** (**a**) Tyrosine 123′ from state2.pdb (derived from 6fqv-no1-one-DNA.pdb) is shown accepting a hydrogen bond from arginine 122′ and also coordinating a catalytic Mn^2+^ ion (at the Y(B) site—the terminal oxygen of the tyrosine is at a similar position to water 5095 in 5cdm—see also Appendix A). The catalytic tyrosine is in the tyrosinate form with a negative charge on the terminal oxygen. (**b**) In the 2.1 Å 2xcs-v2-two-DNA.pdb, a single Mn^2+^ ion is observed coordinated by two waters, Asp 508 and Glu 435 as well as two oxygens from the scissile phosphate. (see Figure 5 in [11]). Water E52 donates hydrogen bonds to the scissile phosphate and the previous phosphate and accepts a hydrogen bond from the NE of Arg 122′. This E52 water is a new feature of the binding pocket—it is some 5.5 Å from Mn^2+^ 5081 (the equivalent water, F41, is some 5.3 Å from Mn^2+^ D5081). Only some hydrogens (white atoms) are shown. (**c**) Superposition of structures from (**a**,**b**). Note the Mn^2+^ ion moves some 3.2 Å between the superposed structures; it remains coordinated by the side chain of Asp 508 (which moves 2.2 Å) and the ‘inside water’ H_2_O 5091 (moves 1.1 Å). Other waters (except E52) and Y123′ (Y123′F) and K581 have been removed from the picture for clarity. Note Arg 122′ also moves—its position in (**a**) is similar position to E52 from (**b**). Only preserved coordination sphere lines (dotted) from OD2 Asp 508 and oxygen of water 5091 are shown to Mn^2+^ ion (purple sphere).

**Figure 4 ijms-26-00033-f004:**
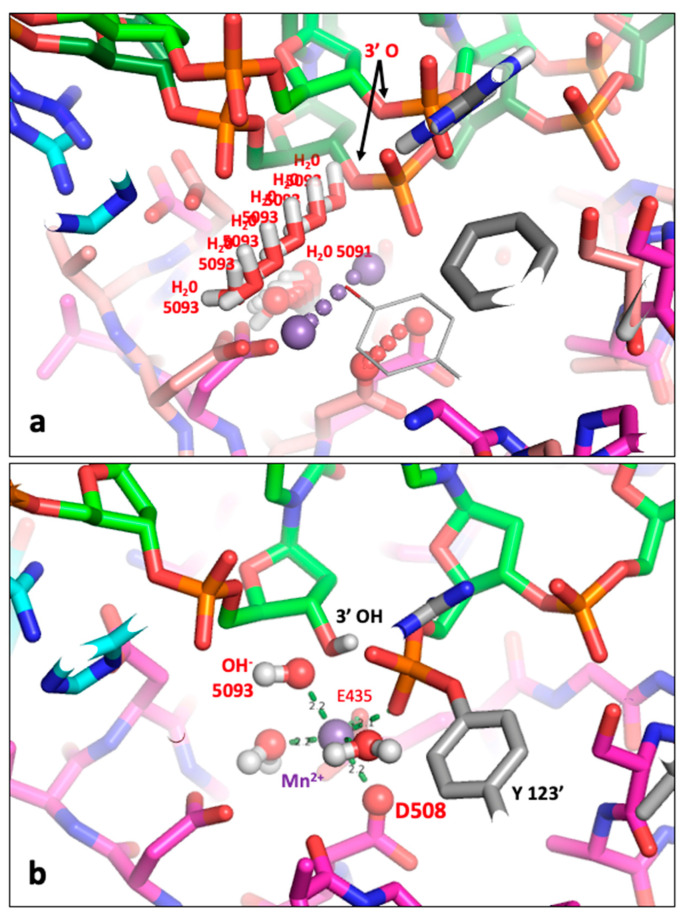
**Does water 5093 protonate the 3′-oxygen to cleave the DNA?** (**a**) Three conserved atoms, the catalytic metal ion (purple), the OD2 from Asp 508 and the oxygen from water 5091 have been used to define movements and suggest that water 5093 moves to protonate the 3′ oxygen to initially cleave the DNA. Some atoms have been removed from the picture for clarity. (**b**) The Mn^2+^ ion (purple sphere) has moved towards the 3′(A) site—thus causing water 5093 to move and protonate the 3′ oxygen from the scissile phosphate, which then becomes the 3′ OH—while water 5093 becomes a hydroxyl ion (OH^−^). In this model, the catalytic tyrosine (Y123′) has just accepted the scissile phosphate—while the catalytic metal ion (Mn^2+^) has an octahedral coordination sphere (two waters, oxygens from Asp B508 and Glu B435 side chains, an oxygen from the scissile phosphate and the oxygen from the OH^−^ 5093 anion). The oxygen atom on D508, which maintains contact with the metal ion throughout the reaction, is shown as a red sphere. Water 5095 is not shown.

**Figure 6 ijms-26-00033-f006:**
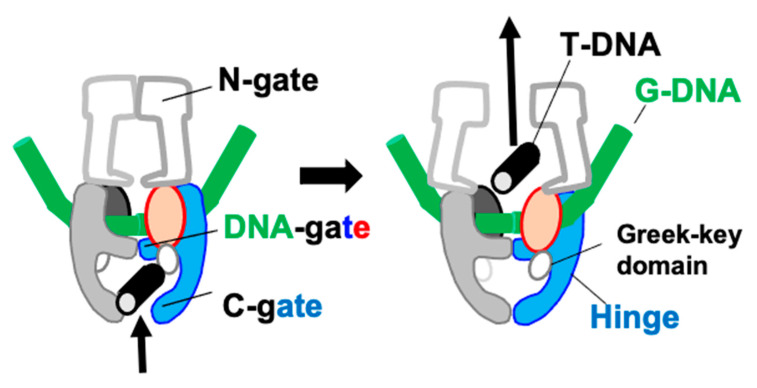
Could the T-DNA move ‘up’ when relaxing positively supercoiled DNA? The C-terminal DNA-wrapping domains are omitted for simplicity.

**Table 1 ijms-26-00033-t001:** Three crystal structures used in this paper (see [10] for details of (re-)refinement of the two metal ion containing P6_1_ crystal structures).

No./PDB Name/Res. DNA Name (Space-Group)	DNA Seq. * −1, +1, +2, +3, +4, +5 +5, +4, +3, +2, +1, −1	Comments/ Catalytic Metals
1./2xcs-v2-BA-x.pdb/2.1 Å 20-20 (P6_1_ a = b = 93.3 Å, c = 412.8 Å)	5′ G G G C C C 3′ 3′ C C C G G G 5′	2.1 Å GSK299423 structure. Y123F uncleaved DNA./ One metal at 3′(A) site at each active site.
2./5cdm-v2-BA-x.pdb/2.5 Å 20-447T (P6_1_ a = b = 93.9 Å, c = 412.5 Å)	5′ C ^Y^ **G G C C G** 3′ 3′ **G C C G G**^Y^ C 5′	2.5 Å QPT-1 structure. DNA cleaved—one compound at each cleavage site./ One metal at Y(B) site at each active site.
3./6fqv-c1.pdb/2.6 Å 20-447T (P2_1_ a = 93.3, b = 124.6, c = 155.2 Å beta = 95.65°)	5′ C G G C C G 3 3′ G C C G G C 5′	2.6 Å binary DNA complex. No DNA cleavage. Two complexes asym. unit./No metal ions observed at active sites.

* Only the central six base pairs of each 20 mer duplex are shown. Conventional numbering relative to the DNA-cleavage site is shown at top (DNA seq.) for each strand. Note complex 2 is cleaved, and the cleaved DNA (underlined) is covalently attached to catalytic tyrosines (^Y^).

**Table 2 ijms-26-00033-t002:** Derived coordinates in preparation for construction of a molecular movie.

Starting Coordinates Filename	DNA Sequence 5′-3′ (- chainID E) 3′-5′ (- chainID F) (Name of DNA)	1st Derived Coordinates 2nd Derived Coordinates	Comments (Note Red Nucleotides Different from Starting Structure)
6fqv-c2.pdb	GAGC GTAC GGCC GTAC GCTT TTCG CATG CCGG CATG CGAG	NA	Coordinates only used for movie. This second complex from the asymmetric unit of 6fqv has different Chain IDs than the other three complexes.
6fqv-c1.pdb	GAGC GTAC GGCC GTAC GCTT TTCG CATG CCGG CATG CGAG	6fqv-c1-one.pdb	Coordinates edited in coot to include Glu B435 + 18 mer DNA sequence: 5′-AGC GTAC GGCC CTAC GGC-3′ 3′-TCG CATG CCGG GATG CCG-5′
2xcs-v2-BA-x.pdb	20-447T *		The inclusion (ordering) of Glu B435 in the catalytic pocket seemed to demand a metal ion (and the movie).
	20-20	2xcs-v2-two-DNA.pdb	Hydrogens added in Maestro (prepare). Waters coordinating catalytic metal energy minimized in Maestro after manual adjustment.
5cdm-v2-BA-x.pdb	GAGC GTAC *GGCC GTAC GCTT* *TTCG CATG CCGG* CATG CGAG	5cdm-v2-one-DNA.pdb	Coordinates edited in coot to single conformations + 18 mer DNA sequence: 5′-AGC GTAC GGCC CTAC GGC-3′ 3′-TCG CATG CCGG GATG CCG-5′
	20-447T *	5cdm-v2-two-DNA.pdb	Hydrogens added in Maestro (prepare: NB delete hydrogens added by default onto Tyr A123, C123, F2009 and E2009 and add bond between Tyr oxygen and phosphate = 1.59 or 1.60 Å). Waters near catalytic metal energy minimized in Maestro

* The same 20-447T DNA was used for the 5cdm and 6fqv crystal structures. However, in 5cdm the protein has cleaved the DNA and the underlined italic *GGCC GTAC GCTT* sequence is covalently attached to the catalytic tyrosine in both subunits.

**Table 3 ijms-26-00033-t003:** Coordinates used in making the molecular movie.

No./Filename	Comments
1. state1.pdb	Derived from 6fqv-c2.pdb. Contains no catalytic metal ion (see Figure 2a).
2. state2.pdb	Derived from 6fqv-c1-one.pdb. Metal ion 5081 and waters 5091, 5093 and 5094 added. The catalytic tyrosine moved slightly so terminal oxygen sits where water 5095 sits in 5cdm (see Figure 3).
3. state3.pdb	Derived from state2.pdb. Moved towards 2xcs-v2-two-DNA.pdb. Water 5095 is based on water E52. DNA not yet cleaved.
4. state4.pdb	Derived from state3.pdb. DNA cleaved as in 5cdm-v2-two-DNA.pdb. Water 5095 has come in from position in state3.pdb.

## Data Availability

Structure factors and ‘original’ coordinates with standard PDB nomenclature are available from the protein databank (PDB). Coordinates with standard BA-x nomenclature are available at https://profiles.cardiff.ac.uk/staff/baxb, (accessed on 12 December 2024)—click on ‘research’ tab—and scroll down to download ‘required’ coordinates. Coordinate files corresponding to the four states used in creation of the molecular movie are available in the Appendix A.

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
