# Peer review of "How Do Gepotidacin and Zoliflodacin Stabilize DNA-Cleavage Complexes with Bacterial Type IIA Topoisomerases? 2. A Single Moving Metal Mechanism"

_ijms, 2024, doi:10.3390/ijms26010033_

Round 1

Reviewer 1 Report (Previous Reviewer 2)

Comments and Suggestions for Authors

The proposed mechanism of DNA cleavage by type IIA topoisomerases, involving the displacement of a single metal, is an intriguing hypothesis, and the authors have provided a detailed explanation and supporting evidence.

Here are some suggestions for improvement: Introduction: The authors should consider shortening the introduction. Although the introductory information is valuable, it is worth simplifying it and focusing on aspects directly related to the proposed new mechanism. It seems that information about the difficulty of the three-body problem in physics and the analogy with topoisomerase mechanisms can be removed. This is not directly relevant to the manuscript.

Discussion: The authors should expand on the topic of alternative mechanisms in the discussion: Briefly discuss the strengths and weaknesses of existing models to better emphasize the novelty of the proposed mechanism.

Author Response

Thank you for your suggestions to aid readability of the Introduction and utility of the Discussion - we have revised the text accordingly.

Reviewer 2 Report (Previous Reviewer 3)

Comments and Suggestions for Authors

The manuscript is suitable for publication.

Author Response

Thank you for approving our manuscript for publication.

This manuscript is a resubmission of an earlier submission. The following is a list of the peer review reports and author responses from that submission.

Round 1

Reviewer 1 Report

Comments and Suggestions for Authors

The authors have presented a very interesting theory regarding the potential DNA cleavage mechanism for DNA gyrase (topoisomerases of type IIA). However, their article entails intrinsic contradictions from its very outset that may exclude its publication.

1. Although the article is labelled as a review, the authors do not present a review of the potential DNA cleavage mechanisms by topoisomerases and related enzymes but instead highlight their theory based on pdb files. If this article was a review, it would be more appropriate to describe the existing data while a minor part could be the opinion of the authors. If the article was of the research type, the authors should have included more data to substantiate their proposal. Right now, the article is more of a research paper with an extended introduction but not a review as its title proclaims.

2. The authors highlight their theory regarding the mechanism based solely on pdb files. This approach, at least as presented in the paper, has some flaws. First, pdb files present a snapshot of the system thus they can potentially support different hypotheses regarding the mechanism of action. Second, the authors have highlighted in the text, the structures with the highest resolution have been mutated so as not to have the Greek Key domain. Although the authors have speculated its function, this domain appears to be a central aspect of the mechanism of action of topoisomerases. Therefore, any hypothesis should consider the specific nature of the domain. Third, the most important drawback of the paper is the lack of computational models regarding the proposed mechanism of action. Molecular dynamics or quantum mechanics simulations can provide valuable insights to this aim. This is of utter importance for the role of water molecules: the pdb files cannot provide this level of information since they do not correspond to potential transition states. Therefore, computational models would potentially bridge this lack of information by providing plausible hypotheses. Changing the position of atoms manually (as described in Methods) to match the proposed hypothesis entails several pitfalls regarding the support of the authors’ theory.

In general, the authors should focus more on the review and less on their proposed mechanism which should be the object of an original research article. A review article should focus on describing the evidence, mainly existing hypotheses besides their own. The authors should also provide an explanation regarding the differences (if any) between the different theories and showcase the evidence. If they aim to highlight their own hypothesis, then the authors should provide a more robust analysis regarding the proposed mechanism.

Other

Line 14. A definition on the species origin of gyrase, or a description where gyrases may be found would be useful for a wider audience.

Fig. 1. Line 110. Please improve the quality/resolution of Fig. 1d. Details are not visible after magnifying the sketch. What do the colorings of figures 1e and 1f signify?

Line 23. “Our model explains why the catalytic tyrosine is in the tyrosinate (negatively charged) form for DNA-cleavage.” What do you mean? This is self-evident. Tyrosine itself could never be a nucleophile.

Line 59. “G-DNA” Do you mean genomic DNA. Please introduce abbreviation. Is it the same in Line 130 (“G-segment DNA”)? Please clarify.

Line 70. “S. aureus” Introduce full name and abbreviation.

Line 83. Introduce here the abbreviation M. tuberculosis for the mentioned Mycobacterium tuberculosis.

Line 84. “A. Baumanii” Introduce full name and abbreviation.

Where is water in Fig. 2? Again, the resolution of the figure must improve as it becomes blurred upon magnification.

Some examples where MD simulations would have given clear answers:

Line 247 “In cleaving the DNA the tyrosinate form is stabilized by interactions with both the catalytic metal (Mn2+) and the positively charged side chain of Arg122 (Figure 3a)”. What is the distance of the OH group of tyrosine from the metal and Arg122?

Fig. 5c. How can tyrosynate be maintained after the contact with Mg2+ has ceased? Could it be possible that water performs a nucleophilic attack on the planar metaphosphate?

Reviewer 2 Report

Comments and Suggestions for Authors

The authors propose a novel mechanism for DNA cleavage by type IIA topoisomerases, based on existing crystal structures. While they acknowledge that their aim is not to definitively prove the mechanism, they present a new explanation that aligns with current data and suggest further experiments for validation. This paper significantly advances our understanding of molecular biology by elucidating the intricate process of DNA cutting by DNA gyrase. Using X-ray crystallography, the authors have determined the three-dimensional structures of DNA gyrase at various stages of the cleavage process. Through computational simulations, they have developed a new method for cutting DNA using a single metal ion. This study provides valuable insights into the molecular workings of DNA gyrase, which could potentially lead to the development of new drugs targeting this enzyme. However, further experimental validation and exploration of the wider biological implications are needed to fully realise the potential of this research. Future studies could include mutagenesis experiments to assess the role of specific aminoacids in the metal ion-DNA interaction, kinetic studies to quantify the effect of the metal ion on cleavage rates, and spectroscopic techniques to directly observe the metal ion-DNA interaction during cleavage. In my opinion, this manuscript, as presented, has real value and should be published. The authors have done a really good job and presented the results well with good discussion. The work is important for the development of molecular biology and will undoubtedly be of great interest to specialists in the field.

Reviewer 3 Report

Comments and Suggestions for Authors

The manuscript is interesting and comprehensive with a deep and profound discussion. The applied methods are clearly detailed, and the study appears to be reasonably planned and executed. In my view, the work represents a scientific level that is above average. However, the only issue that remains unclear to me is why the authors have chosen gepotidacin and zoliflodacin as DNA-cleavage stabilizing agents. I believe this should be elaborated on further in the introduction. As a scientist working in the field of fluoroquinolone studies, I would cite this paper and I recommend its publication.